# Genome Mining of *Pseudanabaena galeata* CCNP1313 Indicates a New Scope in the Search for Antiproliferative and Antiviral Agents

**DOI:** 10.3390/microorganisms12081628

**Published:** 2024-08-09

**Authors:** Michał Grabski, Jan Gawor, Marta Cegłowska, Robert Gromadka, Hanna Mazur-Marzec, Grzegorz Węgrzyn

**Affiliations:** 1Department of Molecular Biology, University of Gdansk, Wita Stwosza 59, 80-308 Gdansk, Poland; michal.grabski@ug.edu.pl; 2Institute of Oceanology, Polish Academy of Sciences, Powstańców Warszawy 55, 81-712 Sopot, Poland; mceglowska@iopan.pl; 3International Centre for Cancer Vaccine Science, University of Gdansk, Kładki 24, 80-822 Gdańsk, Poland; 4DNA Sequencing and Synthesis Facility, Institute of Biochemistry and Biophysics, Polish Academy of Sciences, 02-106 Warsaw, Poland; gaworj@wp.pl (J.G.); robert@ibb.waw.pl (R.G.); 5Department of Marine Biology and Biotechnology, University of Gdansk, Piłsudskiego 46, 81-378 Gdynia, Poland; hanna.mazur-marzec@ug.edu.pl

**Keywords:** cyanobacteria, secondary metabolites, genome analysis

## Abstract

Compounds derived from natural sources pave the way for novel drug development. Cyanobacteria is an ubiquitous phylum found in various habitats. The fitness of those microorganisms, within different biotopes, is partially dependent on secondary metabolite production. Their enhanced production under biotic/abiotic stress factors accounts for better survival rates of cells, and thereby cyanobacteria are as an enticing source of bioactive compounds. Previous studies have shown the potent activity of extracts and fractions from *Pseudanabaena galeata* (Böcher 1949) strain CCNP1313 against cancer cells and viruses. However, active agents remain unknown, as the selected peptides had no effect on the tested cell lines. Here, we present a bottom-up approach, pinpointing key structures involved in secondary metabolite production. Consisting of six replicons, a complete genome sequence of *P. galeata* strain CCNP1313 was found to carry genes for non-ribosomal peptide/polyketide synthetases embedded within chromosome spans (4.9 Mbp) and for a ribosomally synthesized peptide located on one of the plasmids (0.2 Mbp). Elucidation of metabolite synthesis pathways led to prediction of their structure. While none of the synthesis-predicted products were found in mass spectrometry analysis, unexplored synthetases are characterized by structural similarities to those producing potent bioactive compounds.

## 1. Introduction

Cyanobacteria produce a diverse array of bioactive secondary metabolites which help them to survive in competitive environments [1,2]. These compounds are also of interest to researchers focusing on the biotechnological potential of natural products [3,4].

The majority of the discovered cyanometabolites belong to non-ribosomal peptides (NRPs), polyketides (PKs) or hybrid peptides/polyketides (NRPs/PKs) [5]. Biosynthesis of the compounds proceeds according to a thiotemplate mechanism on large multifunctional enzyme complexes of modular structure, recognized as non-ribosomal peptide synthetases (NRPSs) [6], and type I and III polyketide synthases (PKS) [6,7,8]. The enzyme complex sequentially catalyzes the incorporation of proteinogenic or non-proteinogenic amino acids, fatty acids, and other units into the structure of the synthesized molecule. The most frequently identified NRPs include cyanopeptolins, anabaenopeptins, and aeruginoins, while the hepatotoxic microcystins and nodularin represent cyanometabolites biosynthesized through a hybrid NRPS/PKS pathway [6,9]. Cylindrospermopsin, another cyanotoxin, belongs to polyketide alkaloids [10].

Genomic analyses showed the presence of NRPS and PKS gene clusters in 70% of cyanobacterial genomes. However, despite the common occurrence of NRPS- and PKS-encoding genes in cyanobacterial genomes [8,11,12], many products synthesized by these enzymatic complexes are still unknown. Even less is known about the ribosomally synthesized and post-translationally modified cyanopeptides (RiPPs), though genes encoding these peptides are at least equally frequent in the cyanobacterial genome as those encoding NRPSs and PKSs [11]. RiPPs contain only proteinogenic amino acids and are composed of the leader peptide and the core peptide, modified by specific post-translational modification enzymes. This group of peptides mainly includes cyanobactins with thiazole and oxazolin rings [13,14], the tricyclic microviridins [15], and Class II lanthipeptides, characterized by the presence of dehydro-amino acids (dehydrobutyrine—dhb, and dehydroalanine—dha) [16].

In earlier studies, identification of natural products was based on chemical analysis of biological samples. This method is still widely used; however, with the development of modern molecular methods and bioinformatic tools, mining of cyanobacterial genomes has been found to be more effective. Application of these methods revealed that the number of biosynthetic gene clusters (BGCs) in cyanobacteria correlates well with the size of their genomes, i.e., cyanobacteria with larger genomes possess a higher number of BGCs [11,12]. The BGC genes are mainly located on the chromosome, though some have also been found in plasmids [12].

Here, a “bottom-up” approach to cyanometabolite discovery was used to explore the potential of the Baltic cyanobacterium *Pseudanabaena galeata* (Böcher 1949) strain CCNP1313 to produce bioactive metabolites. In previous studies, this strain has shown potent antiproliferative and antiviral activities [17,18]. However, when fourteen peptides isolated from *P. galeata* CCNP1313 and identified based on mass spectra were tested, none of them were found to be active. Therefore, in this work, sequencing and mining of the *P. galeata* CCNP1313 genome were applied to the search for compounds that are potentially responsible for its observed antiproliferative and antiviral effects.

## 2. Materials and Methods

### 2.1. Cultivation

*P. galeata* CCNP1313 isolated in 2010 from the Gulf of Gdansk (Southern Baltic Sea) was classified based on morphological characteristics, genetic sequences (16S-ITS-23S rDNA and *rbcLX;* GenBank accession number MN273769), and ultrastructures [18]. The strain was grown in BG11 liquid medium in 120 mL Erlenmeier flasks at 22 °C (±1 °C) with constant illumination (20 µM photons m^−2^ s^−1^). To obtain an axenic culture, cells were first centrifuged (4000× *g*; 10 min), and the cell pellet was washed and re-suspended in fresh sterile BG-11 medium; the procedure was repeated three times. The strain was then further purified by transferring the cell suspension onto a solid (1.5% agar) BG11 medium, where part of the Petri dish was darkened to allow for phototaxis. The cells were then picked and transferred to a liquid BG11 medium and incubated for two weeks. As a final step, *P. galeata* CCNP1313 was inoculated into a BG11 medium supplemented with ceftriaxone (100 µg mL^−1^; Pol-Aura, Olsztyn, Poland). The axenity of the culture was checked by inoculation on Luria–Bertani agar (1.5%) plates. The cyanobacteria were then inoculated into a fresh BG11 medium and incubated for three weeks, after which the cells were centrifuged and the biomass was frozen for further testing.

### 2.2. Isolation and Sequencing of Genomic DNA

Genomic DNA of bacterial strains was isolated using the SDS/phenol method, as described previously [19,20]. DNA quality control was performed by measuring absorbance at 260/230 nm using PicoDrop (Picodrop Ltd., Cambridge, UK), and template concentration was determined using a Qubit fluorimeter (Thermo Fisher Scientific, Waltham, MA, USA). DNA integrity was analyzed by 0.8% agarose gel electrophoresis and by capillary PFGE using a gDNA 165 kb kit and FemtoPulse instrument (Agilent, Santa Clara, CA, USA).

An Illumina-compatible paired-end sequencing library was constructed using the NEB Ultra II FS Preparation Kit (New England Biolabs, Beverly, MA, USA) according to the manufacturer’s instructions. This library was sequenced using an Illumina MiSeq platform (Illumina, San Diego, CA, USA) with 2 × 300 paired-end reads using a v3 600-cycle sequencing kit. Sequence quality metrics were assessed using FASTQC (http://www.bioinformatics.babraham.ac.uk/projects/fastqc/ accessed on 16 February 2023) and quality-trimmed using fastp [21].

Prior to long-read library preparation, genomic DNA was sheared into ~30 kb fragments using 26G needle followed by size selection using a Short Read Eliminator kit (Circulomics, Baltimore, MD, USA). Then, 5 µg of recovered DNA was taken for 1D library construction using a SQK-LSK109 kit, and 0.8 µg of the final library was loaded into an R9.4.1 flowcell and sequenced on a GridION sequencer (Oxford Nanopore Technologies, Oxford, UK).

### 2.3. Genome Assembling

Raw nanopore data were base-called using Guppy v5.0.7 in super accuracy mode (Oxford Nanopore Technologies, Oxford, UK). After quality filtering using NanoFilt [22] and residual adapter removal using Porechop (https://github.com/rrwick/Porechop accessed on 16 February 2023), the obtained dataset was quality checked using NanoPlot [22]. Long read assembly was performed using the Dragonflye pipeline (https://github.com/rpetit3/dragonflye accessed on 16 February 2023). In brief, nanopore reads were assembled using flye in nano-hq mode followed by polishing with racon and medaka. Long-read-assembled contigs were further polished using Polypolish (https://github.com/rrwick/Polypolish accessed on 16 February 2023) and POLCA [23]. Genome binning and taxonomic classification was performed using MaxBin (v2.2.7) and Kraken2 with the Kraken standard library, accordingly.

### 2.4. Genome Annotation and Analysis

The assembled genome was annotated using the NCBI Prokaryotic Genome Annotation Pipeline (PGAP). Open reading frames (ORFs) were verified by Prodigal and further submitted to antiSMASH (v6.1.1), SeMPI (v2.0), ClusterMine360, and NaPDoS v2.0 to obtain regions encoding secondary metabolite synthetases. Sequences were aligned using Clustal Omega, nucleotide–nucleotide and protein–protein BLAST v2.12.0+ [24,25] against sequences obtained from the NCBI, PDB, and AlphaFold databases [26,27,28]. Operon-mapper was used to predict operons in the *P. galeata* CCNP1313 genome [29]. The search for peptide motifs within coding sequences was performed using SeqKit v2.5.1’s locate subcommand [30]. Selected regions were subjected to the prediction of evolutionarily-conserved protein domains and a motif search against the Conserved Domain Database v3.20 [31]. The secondary structures of selected proteins were elucidated by Jnet v.2.3.1 and I-Tasser v.5.1 [32,33]. Transmembrane proteins were predicted using TMHMM v1.0.24. Stem-loop structures in DNA sequences were found using the DNA Punctuation program (https://github.com/mariapoptsova/dnapunctuation accessed on 16 February 2023). Manipulation of the defined sequences was carried out using samtools [34]. Sequence alignment was visualized by BOXSHADE v.3.31. The genome was visualized using the CGView Comparison Tool (CCT) [35].

### 2.5. Genomic Data

The whole genome sequence of *P. galeata* CCNP1313 has been deposited in GenBank within the BioProject no. PRJNA884271. The statistics of the annotation are provided in Appendix A.

### 2.6. LC-MS/MS Analysis

Lyophilized biomass of CCNP1313 (20 mg) was extracted with 75% methanol in water and analyzed by LC-MS/MS. The LC was coupled to a hybrid triple quadrupole/linear ion trap mass spectrometer QTRAP5500 (Sciex, Toronto, ON, Canada). Chromatographic separation was achieved with a Kinetex C18 column (100 × 4.6 mm, 2.6 µm, 100 Å) (Phenomenex) using a water–acetonitrile mixture (and both solvents having 0.1% formic acid) as a mobile phase (0.4 mL/min). Ionization was performed with a Turbo Spray ion source in positive mode. The spray voltage was maintained at 5.5 kV, 550 °C, curtain gas pressure was 20 psi, nebulizer gas pressure was 55 psi, and declustering potential was 80 eV. Full-scan mass spectra were acquired over a range from *m*/*z* 100 to 1250. For detection of ions, the Information-Dependent Acquisition mode (IDA) was used with a collision energy of 60 eV.

## 3. Results and Discussion

### 3.1. Analysis of the P. galeata CCNP1313 Genome

The genome of *P. galeata* CCNP1313 consists of six replicons, a circular chromosome of 4,928,719 bps, and five circular megaplasmids (Figure 1).

The total genome size is 5,842,326 bps. Genome annotation of all replicons distinguished 5320 genes, with 5176 loci potentially encoding proteins (CDSs) and 54 encoding functional RNAs. A total of 90 loci were deemed pseudogenes.

The genome sequence of *P. galeata* CCNP1313 was searched for potential secondary metabolite biosynthetic gene clusters, resulting in recognition of one gene producing amino acid condensation domain-containing protein (chr position 842,738 bp–848,680 bp), and the gene cluster (chr position 1,530,477 bp–1,548,363 bp), which besides the previously mentioned domain, which is specific to the NRPS system, contains ketosynthase (KS), revealing a hybrid NRPS-PKS system. Moreover, bearing condensation domains, standalone genes coding for β-ketoacyl-acyl-carrier-protein synthetases were found within the chromosome at positions 4,506,072 bp–4,507,325 bp and 4,767,642 bp–4,768,802 bp, both putatively engaged in fatty acid biosynthesis. Additionally, a type III PKS-encoding sequence had been found within chromosome spans at positions 2,381,211 bp–2,382,323 bp. The biological activity of this cyanobacterial strain, as described previously [18], was linked to peptides; however, none of the above-mentioned NRPS systems contained enough modules to sustain production of peptides with the respective length. Nor were ribosomally synthesized peptides derived from type II lanthipeptide synthetase, the encoding sequence of which, unlike other such genes, was located within plasmid spans (pPg_03 position 9308 bp–12,523 bp). Thus, peptide sequences found previously [18] were utilized for the search for peptides within coding sequences of the *P. galeata* genome.

All of the considered synthases were previously found in various *Pseudanabaena* species, but despite their genome structures being deposited, none of them were published. A gene encoding a non-ribosomal peptide synthase OA858_03905 was reported in the *Pseudanabaena frigida* (Fritsch 1912) Anagnostidis 2001 metagenome assembled genome (MAG) (PZO39803.1) and *Pseudanabaena* sp. FACHB-1050. As for the PKS/NRPS synthase-encoding region, a cluster of genes encoding potential hybrid synthase were found in *Pseudoanabaena* sp. SR411, Iw0831 and BC1403. The PKS synthase gene sequence was found in MAG of *Pseudanabaena* sp. (PZU95803.1). The RIPP synthase genes were also found in *Pseudanabaena* BC1403 and FACHB-1050 genomes. Type I/II fatty acid synthases are highly conserved and were found in all the above-mentioned genomes and many others (27 cyanobacteria genomes with >90% protein identity), as they are key components in the elongation of fatty acids. As previously mentioned, the differences in substrate affinity between FabB and FabF on the FAS pathway are resolved and discussed elsewhere [36].

### 3.2. Non-Ribosomal Peptide Synthase (NRPS) Gene

Annotation of the selected region revealed one open reading frame (ORF) (OA858_03905) composed of 5943 bp transcribed in the reverse direction (Figure 2).

The gene product is composed of two repetitive modules, each responsible for the incorporation of a single residue into an elongating peptide, in which case the studied synthase is capable of producing dipeptides. The first module consist of two domains, namely adenylation (A) and the peptidyl carrier protein domain (PCP), facilitating delivery of amino acid to the condensation domain (C) of the second module. The subsequent module, apart from the above-mentioned condensation domain, contains an adenylation domain, a peptidyl carrier protein domain, and a thioesterase domain, located on the C terminus of the potential protein. Based on prediction of L-amino acid substrates for the adenylation domain [37,38], the analyses led to an inconclusive structure of the A domain binding pocket residues, as the first module of NRPS holds serine residue in position 235, which is atypical of α-amino groups (corresponding to the position numbering in GrsA Phe), instead of aspartic acid. Asp in this position is essential for mediating electrostatic interactions with the α-amino group of the bound amino acid (Table 1).

However, lysine within a strictly invariant C-terminal residue 517, which is responsible for binding with the carboxylate group and the ribose moiety of adenylate, remained unchanged. All amino acid positions accommodating a substrate in the first A module were found to be identical within three other cyanobacteria bearing similar synthases: PZO39803.1, MBW4579359.1, and MCL6435924.1 (Figure 3).

The lack of Asp in A domains indicate that they do not have an α-amino group; thus, a specificity-conferring code may not be applicable here as those domains may harbor unusual substrates [37]. However, the A domain harboring serine in residue 235 was found in EntE mutant N235S, which could bind 2-aminobenzoic acid (2-ABA) [39], and also in the product of the dhbE gene (UniProt Q840D1) (a ligase), potentially binding 2,3-dihydroxybenzoic acid (DHB), which was also a substrate for EntE prior mutation [40]. Pairwise alignment of OA858_03905 revealed four different residues within the substrate-binding pocket of the second module, where instead of Arg, binding pockets of mentioned cyanobacterial synthases recognize 3h4mPhe. Arginine (Arg) substrate recognized by the second module A domain was then loaded by PCP onto the terminal type-I thioesterase domain (TE) active site, releasing the product from NRPSs.

Substitution in module 1 of the NRPS A domain residue Asp235 is recognized in hydroxy/carboxy-acid-activating domains [37]. A condensation reaction between hydroxy acid and amino acid may produce depsipeptides, compounds with both ester and amide bonds (reviewed in [41]). This bond’s formation may be facilitated by the condensation domain of the *P. galeata* CCNP1313 NRPS A domain of the first module, while Ser235 residue may enable the binding of hydroxy acid to the first module of NRPS. Whether or not the product harbors a non-amino acid moiety, the two-module structure of NRPS can act as a dipeptide producer. Derivatives of dipeptides as well as depsipeptides have been proven to possess a broad spectrum of biological activities [42,43,44].

### 3.3. The Mixed PKS/NRPS System

The potential PKS/NRPS synthetase-encoding region consists of three ORFs (OA858_07105, OA858_07095, OA858_07090) bearing core biosynthesis modules (Figure 4).

Located on the reverse strand, ORFs OA858_07105 and OA858_07095 are intersected by 2557 bp long span bearing additional ORF (OA858_07100) where no domains were detected. The first gene (OA858_07105) codes for a protein containing a minimal NRPS-specific chain elongation module, with an additional PKS module and PKS-specific acyl transferase domain. The protein holds a condensation (C) domain at the N-terminus, followed by the adenylation (A) domain, the PCP domain, the PKS domain, and the PCP domain. Lastly, PCP transfers a nascent product to C-terminal acetyl ornithine like the aminotransferase domain, which offloads the emerging product to the first module of ORF OA858_07095. ORF OA858_07095 contains a conserved HHxxxD motive in the C domain, found at the N terminus, which suggests that this domain may be active. We consider this gene to be a part of the hypothetical PKS/NRPS synthase-encoding region, as the first C domain active site is typically inactive. This gene contains every module (C, A, PCP) necessary to encode a protein section capable of incorporating amino acid into a nascent peptide–polyketide hybrid. The stop codon of the second gene overlaps with the start codon of the third ORF (OA858_07090), which indicates that those genes could be transcribed as an operon (97% probability). The third gene of this cluster encodes a protein which encompasses the N-terminal docking domain and the thioesterase domain at the C terminus, on top of standard condensation, adenylation and phosphopantetheine attachment site found in NRPS modules.

The protein sequence of the first module encoded by ORF OA858_07105 architecturally resembles the surfactin A synthetase C NRPS; thus, unlike SrfC (PDB 2vsq), the adenylation domain of the first NRPS-like module putatively activates the valine (Val) substrate (Table 2).

Activated amino acid is offloaded by PCP, which transfers the bound amino acid intermediate to a subsequent module, more specifically the β-ketosynthase (KS) subdomain of the PKS module. Unique PKS in comparison to typical PKS I (PDB 2HG4) lacks the AT domain, which might be a feature of amino acid accepting KS [45]. KS mediates C-C bond formation between the aminoacyl chain and malonyl-type extender, selected by acyltransferase (AT), which is bound to the acyl carrier protein (ACP) found closer to the C-termini of the protein. The structure of the PKS-containing module, encoded in ORF OA858_07105, resembles 6-Deoxyerythronolide B synthase (DEBS) module 1 (PDB 7m7f). Each module of DEBS adds a single (2S)-methylmalonate monomer to the nascent chain, extending it by one extender unit [46], which suggests that the OA858_07105 PKS domain may act in a similar manner. The bound hybrid chain is then transaminated by the pyridoxal 5′-phosphate-dependent (PLP) aminotransferase III domain found at the C-terminal of OA858_07105 encoded protein, which is able to catalyze transamination reactions between amino acid (donor) and keto acid (acceptor) [47]. Transamination conducted by the aminotransferase domain can introduce an amino group at the β-position of polyketides to form a β-aminoacyl-ACP intermediate. Release of this intermediate depends on condensation with another aminoacyl-ACP which may be derived from a subsequent module found within the protein encoded by OA858_07095 ORF [48]. The condensation domain of the OA858_07095 ORF transfers the intermediate from the upstream ACP domain, encoded by ORF OA858_07105, to an amino moiety of the substrate bound by the PCP domain of the OA858_07095 protein. PCP belongs to the first module in which the binding pocket of the A domain activates Val/Trp or Phe. In a similar fashion as previously described, the nascent product is transferred by PCP to the C domain encoded by ORF OA858_07090. The adenylation domain of the last NRPS/PKS module, encoded by ORF OA858_07090, activates the Thr/Val substrate. After addition of this moiety, the complete product is offloaded by the terminal TE, found at the C terminus of the encoded protein.

It has been proposed that the aminotransferase (AMT) domain in the NRPS/PKS hybrid enzyme, located between PKS and the NRPS module, is responsible for the conversion of fatty acid (activated by the PKS domain) into an amino acid [49,50]. Thus, activation of acyl substrate by the PKS domain encoded upstream of aminotransferase has a profound effect on the structure of amino acid. Most commonly, the extender units are malonyl-CoA and methylomalonyl-CoA, yet modular PKSs could be naturally specific for certain α-carboxyacyl-CoA [51]. Transfer of the α-carboxyacyl-CoA extender unit to ACP occurs via the acyltransferase domain of the PKS module, where a condensation reaction is catalyzed by KS-form β-ketoacyl-ACP [52]. Promiscuity in acceptance of aminoacyl substrates by KS domains had been previously noted [45]. The catalytic triad of KS subdomains in the *P. galeata* CCNP1313 NRPS/PKS protein sequence harbors Cys (1056), His (1192), His (1232) residues, much like FAB or PKS I KS, which facilitates elongation of the product by subsequent transacylation and condensation reactions. In the transacylation step, the transfer of the putative Val form upstream of module 1 via ACP to the Cys (1056) residue of KS is proceeded by condensation, facilitated by two His residues, between Val and the malonyl-CoA unit bound to AT. The substrate specificity of the AT domain was elucidated on the basis of GHSI and HAFH motifs present in OA858_07105 in AT positions of the putative protein [53,54], due to the PCP transfer of the nascent chain to the glutamate-1-semialdehyde aminotransferase domain installing an amino group at the β position of the acyl chain. The addition of an amino group at the β-position of poliketides has previously been noted in microcystin-type cyclic peptides (reviewed in [48]). Nodularin (NdaF subunit), mycosubtilin (MycA), and microginins (micD) all contain a β-amino fatty acid moiety added via AMT and encoded in between PKS and condensation domain sequences, as similarly found in OA858_07105 (Moffitt and Neilan) [49,55,56]. However, in the biosynthesis of β-amino acid of nodularin (Adda), four PKS modules are involved in extending the phenylacetate starter unit by four malonate moieties [50]. While solely based on nucleotide sequencing, we cannot rule out whether PKS I in this synthetase is iterative or non-iterative, as the PKS module does not contain a dehydration subdomain typical of iPKS. Moreover, the sheer size of NRPS/PKS gene cluster encoding the synthetase of *P. galeata* CCNP1313 (18.2 kb), which is incomparably smaller than the others primarily derived from NRPS hybrid enzymes, should exclude condensation of such a product (reviewed in [6]). Our findings suggest that a β-amino fatty acid moiety may be inserted between first two residues within the Val-Val/Trp/Phe-Thr/Val tetrapeptide.

### 3.4. Polyketide Synthase (PKS) Gene Clusters

A standalone type III polyketide synthetase-encoding gene was found within a range of 2,381,211–2,382,323 bp of ORF OA858_10790, presenting domain architecture similar to chalcone (CHS)/stilbene (STS) synthases. However, based on identity of amino acid sequences of OA858_10790, only 26–29% of identities matched CHS/STS synthase proteins (UniProt: P30073, P30073, P23418; PDB 1EE0), among which amino acid identity was above 70%. Catalytic triad residues in the OA858_10790 product are typical of CHS-like KAS III or FabH enzymes, which use malonyl-CoA as an extender unit [57]. A putative protein encoded by OA858_10790 possesses Cys in its active site (155), to which the acyl-enzyme chain is bound during elongation reactions, while decarboxylation of malonyl-CoA during elongation is catalyzed by His (291) and Asn (324) (Jez and Noel, 2000). However, amino acid residues in positions T197, G257 and S338 based on CHS numbering differ. Alternation by substitution of those CHS residues, to correspond to the 2-PS active site, resulted in functional conversion of CHS to 2-PS, thus changing the starter substrate and shifting the resulting chain length [58,59]. Aligned within the positions H (188), I (246), and S (327), amino acids of the OA858_10790 product were found to match the Type III PKS1 in the brown algae *Ectocarpus siliculosus * (Dillwyn) Lyngbye, 1819 (40% overall protein identity). Able to utilize several substrates as the starter molecule instead of p-coumaroyl-CoA as in CHS, the structure of the PKS1 biosynthetic products was dependent on the starter unit of an acyl chain; thus, the produced compound of OA858_10790 could not be elucidated solely based on its amino acid sequence [60].

### 3.5. Fatty Acid Synthases

The first step in fatty acid synthesis turns acetyl-CoA into malonyl-CoA via a biotin-dependent carboxylation process, which is conducted by multi-subunit acetyl-CoA carboxylase (ACC) encoded by four genes *accA* (OA858_18170), *accB* (OA858_01765), *accC* (OA858_03000), and *accD* (OA858_17745). A transfer of the malonyl group from CoA to ACP, encoded by the *acpP* gene, is conducted by the enzyme encoded by *fabD* (OA858_18515). When the acyl chain is attached to ACP, it enters the elongation cycle, conducted by FabB and/or FabF enzyme(s). The former ORF, encoding β-ketoacyl-ACP synthase (OA858_20590), was found to bear a *fabF*-specific domain. As it has 75% identity with the encoded protein, it is similar to β-ketoacyl-ACP synthase type II (SMTL ID 1e5m.1). This condensation enzyme is able to carry out subsequent rounds of fatty acid elongation reactions, extending the chain by binding acyl chain intermediates, covalently bound to ACP and encoded 63 nt upstream (OA858_20585; 65% identity with malonyl-ACP SMTL ID 2x2b.1) [61,62]. The latter ORF also encodes β-ketoacyl-ACP synthase, while unlike OA858_20590, OA858_21775, it has a FabB-specific domain. In both of those synthases, the catalytic triad consists of Cys/His/His. The condensation product is then reduced to β-hydroxyacyl-ACP by FabG (OA858_14210), which then is dehydrated by FabZ, encoded by OA858_12125 CDS. The subsequent dehydration reaction produces enoyl-ACP, which can be fully reduced by FabI (OA858_03590), thus completing the synthesis of the acyl chain.

### 3.6. Ribosomally Synthesised and Post-Translationally Modified Peptides (RiPPs)

The putative lanthipeptide synthetase LanM-like is encoded by ORF (OA858_24670), which was found to be located in the plasmid pPg_03 reverse strand. Upstream of the putative synthetase-encoding sequence, an OA858_24665 ORF was found, coding for a precursor peptide (Figure 5).

The RiPP precursor peptide can be divided into the N-terminal leader peptide and the C-terminal core peptide of lanthipeptide, as a double glycine motif was found within its sequence. The motif indicates a cleavage site, downstream of which a putative lanthipeptide precursor was found at the C-terminal region of the RiPP precursor (Figure 5). The N-terminal leader sequence holds the FXXXV motif, shown to be critical for binding of the leader peptide to the synthetase [62,63]. The C-terminal core peptide is processed by class II lanthipeptide bifunctional synthetase, which performs dehydration and cyclization by the N-terminal dehydratase domain and the C-terminal cyclase domain, respectively. However, unlike class II lanthipeptide, the protein sequence of OA858_24670 ORF lacks signature zinc ligands (His/Cys/Cys residues) in the cyclase domain, which are hypothesized to activate Cys thiol in the cyclization step [64,65]. This finding is in agreement with the translated sequence of OA858_24665 ORF, encoding the precursor peptide, as it lacks cysteine residues. Nonetheless, Ser (2) and Thr (1) residues, susceptible to dehydration, were present in the translated sequence.

At the position placed 8991 bp downstream of the putative RiPP synthetase-encoding locus, three ORFs were found located on the forward strand, coding for proteins putatively responsible for secretion (OA858_24710) and ABC-type transport (OA858_24715, OA858_24720) of bacteriocins/lantibiotics. They all belong to a defined three-gene putative bacteriocin transport operon, whereas OA858_24715 contains a region encoding a six-transmembrane helical domain (6-TMD) of the ABC subunit of the bacteriocin system. More importantly, at the N-terminus of this protein, a peptidase domain was found that cleaves a double glycine (GG) motif-containing signal peptide from lantibiotic and non-lantibiotic bacteriocins before secretion [66] (Figure 6).

The OA858_24665 gene product belongs to Nif11-related peptide (N11P) family precursors, according to the TIGR03798 domain found within leader sequence. N11P members, as in this case, are encoded adjacent to LanM-like lanthionine-forming encoding enzymes and a gene cluster of ABC-transport-encoding proteins [67]. The amino acid sequence of this precursor peptide has a longer leader than previously described for this family (83) and these suffix (29) sequences. The leader peptide, encoded by OA858_24665 gene contains the E(-8)LXXVXGG(-1) motif typical of class II lantibiotics [63]. However, as suggested by Welker [68], a lack of Cys in the core peptide would exclude the ORF encoded by OA858_24665 as a potential lanthipeptide precursor because its active molecule would lack lanthionine. Cyanotins is the name suggested for this family of RiPPs, which are highly divergent in their core peptide sequence.

Different amino acids within the crucial zinc binding site of A875C, M911C, and W912H according to CylM numbering suggest an inability to conduct cyclization by the putative LanM of *P. galeata* CCNP1313. Thus, we can surmise that with no Cys residue, a core peptide encoded downstream of putative LanM is a non-lanthibiotic. In LctM, mutation within zinc ligands impaired cyclization activity but had no effect on dehydration of peptides; however, its linear product lacks antimicrobial activity [69]. Sequence alignment of putative LanM synthetase from *P. galeata* CCNP1313 to CylM (with 21.62% sequence identity) revealed an activation loop, a residue segment forming a part responsible for polypeptide binding and subsequent formation of dehydro-amino acids from Ser/Thr residues [70]. The loop in protein encoded by OA858_24670 is formed in between Asp397 found within the sheet and Glu418 at the helix, a segment architecturally resembling lipid kinases [71]. Asp in this position of synthetase was found crucial as its mutation lead to an incapability to eliminate phosphate from phosphorylated peptide, hence abolishing its dehydration activity [69]. The core peptide of *P. galeata* CCNP1313 is hydrophobic (62.07% of hydrophobic residues), with two residues, at the N-(acidic) and C-terminus (basic), which are susceptible to modifications. The N-terminus Glu amino acid is prone to spontaneous cyclization resulting in pyroglutamine formation, while at the C-terminus, oxidation may occur due to Trp residue, which could introduce structural changes into peptides. Moreover, Trp placed at the C-terminus of hydrophobic peptides was proven to enhance hemolytic and cytotoxic activity [72]. Lysine is the only basic amino acid occupying the (-1) residue from the C-terminus. Feasible transformation within the core peptide may engage Lysine in bridge formation with Dha. Cross-linking between those moieties creates lysinoalanine (Lal), the formation of which can be induced by increasing pH or temperature, thus impairing access to peptides for proteolytic enzymes, all without enzymatic activity (Friedman 1999 [73]). While the mechanism of forming Lal is not fully understood, structural analogs of enzymes engaged in the generation of Lal, as in duramycin (durN) or cinnamycin (cinN) lanthibiotics, were not found in the CCNP1313 LanM-like cluster [74,75].

### 3.7. Peptides Previously Found in Mass Spectrometry (MS) Analysis

According to the analysis of the *P. galeata* genome, peptides described previously [18] could not be derived from the known NRPS synthetases, as they do not contain enough modules to carry out biosynthesis of non-ribosomal peptides of such mass. Moreover, it appears impossible to synthesize those products using the above-described RiPP synthetase. However, using amino acid sequences of peptides described in the above-mentioned paper [18], a pairwise alignment was deployed in the search for short amino acid spans within translated coding sequences of the *P. galeata* CCNP1313 genome. This analysis resulted in recognition of four out of fourteen structures of peptides, namely PG638, GP655, GP767, and GP598 (Table 3). Due to the use of the liquid chromatography–tandem mass spectrometry (LC-MS/MS) method, which does not allow us to discriminate between the isobaric ions of Leu and Ile, a regular expression was applied within those positions, accounting for the described discrepancies. As a result, single peptides matched within 11 proteins; thus, 5 of them were buried within secondary structures of membrane-bound proteins, and the rest were deemed globular (Table 3). All five of those membrane-bound proteins were predicted to be transmembrane, although only the GP767 peptide was found outside the cell membrane bound to the OA858_19390 photosystem I core protein PsaB.

### 3.8. Cyanophage DNA Sequence in the P. galeata CCNP1313 Genome

We have taken into consideration the possibility that the issue with identifying potent antiproliferative and antiviral agents in the *P. galeata* CCNP1313 strain might be related to cyanophages, particularly prophages present in the cyanobacterial genome, which could encode proteins hindering the desired activities. Therefore, we have analyzed the genome of the investigated strain for the presence of prophage-derived DNA sequences. Indeed, using PHASTER, a phage search tool [76], we have identified incomplete prophage sequences within the genome. Putative prophage sequences were found within the chromosome, between 707,842 bp and 727,624 bp nucleotides. This region was homologous to several phages, of which Planktothrix phage PaV-LD (NCBI: NC_016564.1), a cyanophage with 95,299 bp long genome, held the highest number of common genes (three). Out of 13 phage-related genes within the respective 19,783 bp span (21 genes in total), endonuclease-, transposase- and protease-encoding genes have been found. The rest of the phage-like genes were deemed phage-like proteins or hypothetical proteins. No morphogenesis-related genes were found. We conclude that the region contains insufficient information to elucidate on cyanophage activity within the *P. galeata* CCNP1313 genome. Therefore, the hypothesis on the interferences of phage-derived products with the activities of identified or predicted compounds seems to be unlikely.

## 4. Concluding Remarks

Silent genes of synthesis clusters or insufficient yield of product obtained from cyanobacteria producers are an apparent problem encountered in unraveling the potential of secondary metabolites [77,78]. Herein, a “bottom-up” approach led to the distinction of four classes of newly described synthetases putatively producing secondary metabolites in *P. anabaena* CCNP1313. While products solely based on nucleotide sequencing may only merely resemble actual synthesis products, such insight is crucial for selection and further handling of biosynthetic pathways of potent metabolites. The outlined pathways unraveled the desirable features of NRPS/PKS hybrid synthetases in heterologous expression due to the sheer size of the cluster. However, a lack of homologous hybrid synthetases exposed a gap in our knowledge, which led to the uncertain structure of the predicted peptide. The same goes for the NRPS cluster, in which the first module incorporated hydroxy or carboxy acid to which GrsA Phe numbering does not apply. Lanthipeptides, on the other hand, were more frequently analyzed by genetic methods than detected by LC-MS/MS, in fact lanthipeptides have been detected only in *Synechococcus* and *Prochlorococcus*. However, prior to genomic investigation, due to the sheer size of the core peptide and poor solubility fractions, it would have been omitted, stressing the urgent need to interlace different methods. While none of the predicted products of the *P. galeata* CCNP1313 genome were detected via LC-MS/MS, being hampered either by their specific features (e.g., ionization conditions) or their *m*/*z* value outside the detection range of the applied instrument, the putative compounds remain unknown. Therefore, one might propose that in further studies, it is necessary to use conditions that ensure the higher stability of tested compounds or their protection against destabilizing factors. Moreover, using an MS system that works for a higher range of molecular weights is recommended.

Another intriguing phenomenon is that among 14 peptides isolated from *P. galeata* CCNP1313, none were active. One possibility is that activity cannot be attributed to isolated peptides (tested alone) but rather to a suitable combination of peptides. If this is true, one might even speculate that some of candidates for such compounds, acting together with others, could belong to the group of molecules identified in this work on the basis of in silico genome analysis.

## Figures and Tables

**Figure 1 microorganisms-12-01628-f001:**
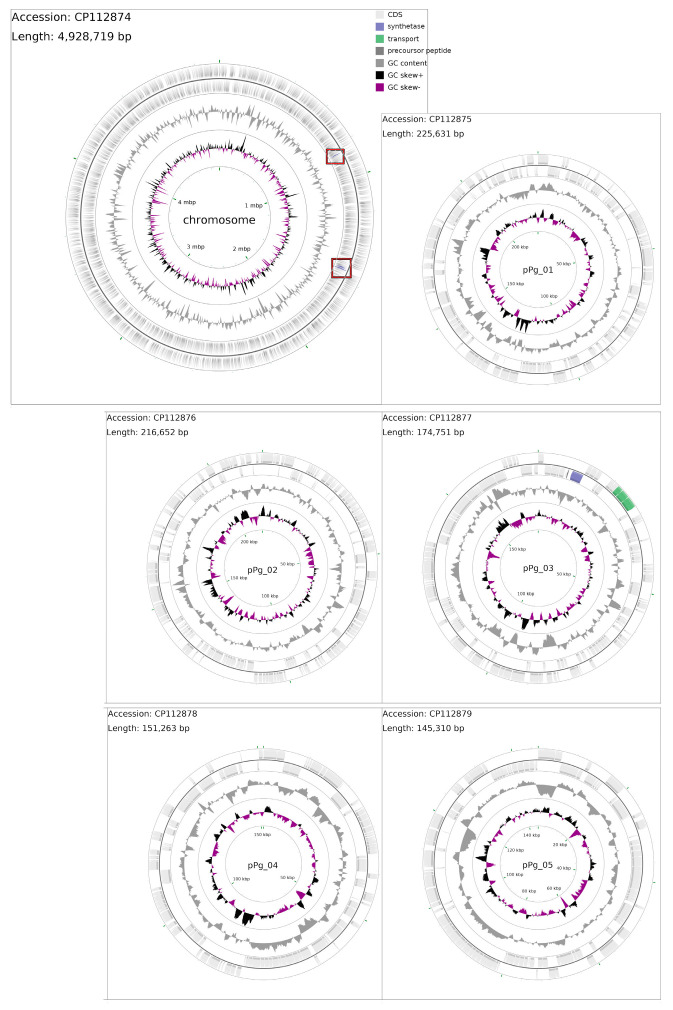
Whole genome of *P. galeata* CCNP1313, with a distinguished chromosome and five plasmids (pPg_01-05). The two outermost rings show coding sequences, where the outer ring shows CDS on a direct strand and the inner ring holds CDS encoded on a reverse strand. The middle circle shows GC content (gray), and the innermost circle shows GC skew (black and purple). All hypothetical peptide synthetases were marked blue within reverse strands on the chromosome and pPg_03.

**Figure 2 microorganisms-12-01628-f002:**
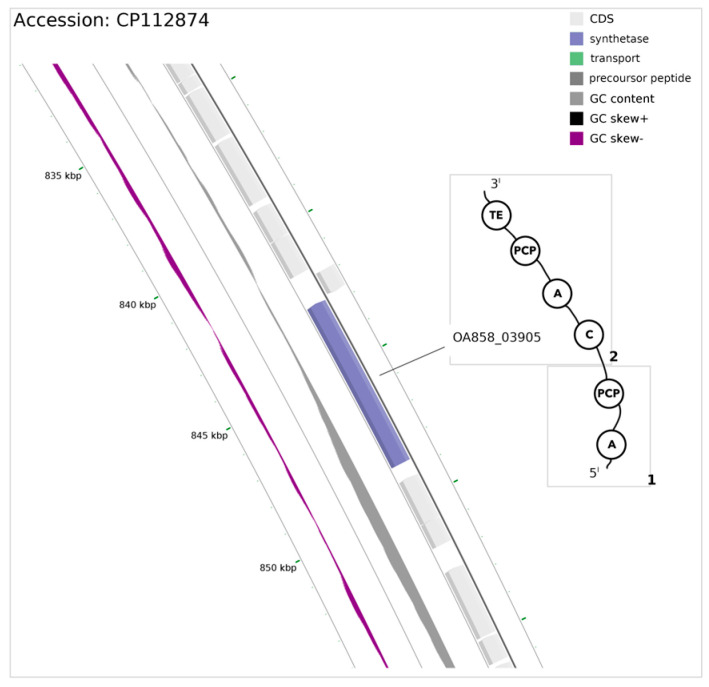
Closeup of chromosome region encoding NRPS synthetase in the reverse direction, within a range of 842,738-848,680 bp, marked blue. Circles represent domain organization within dipeptides producing nonribosomal synthetase. Domains are abbreviated as follows: A, adenylation; C, condensation; PCP, peptidyl carrier protein; TE, thioesterase. Domains in rectangles (1, 2 in bold) correspond to module numbering in Table 1.

**Figure 3 microorganisms-12-01628-f003:**
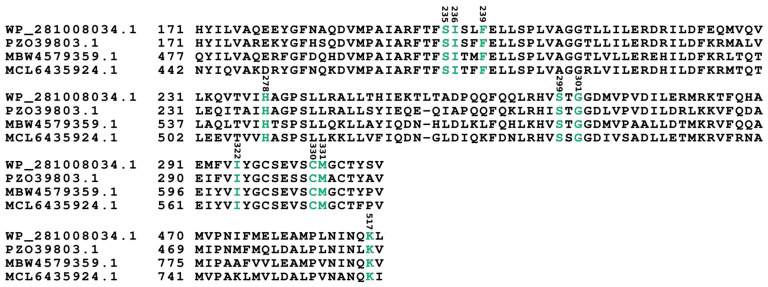
Alignment of the first module adenylation domain from *P. galeata* CCNP1313 synthetase (WP_281008034.1) and three other cyanobacteria-derived NRPSs. The residue positions indicated in the above amino acid sequences were set according to GrsA Phe numbering. Identical amino acids which are crucial for the activity and occuring in all analyzed proteins are marked in green.

**Figure 4 microorganisms-12-01628-f004:**
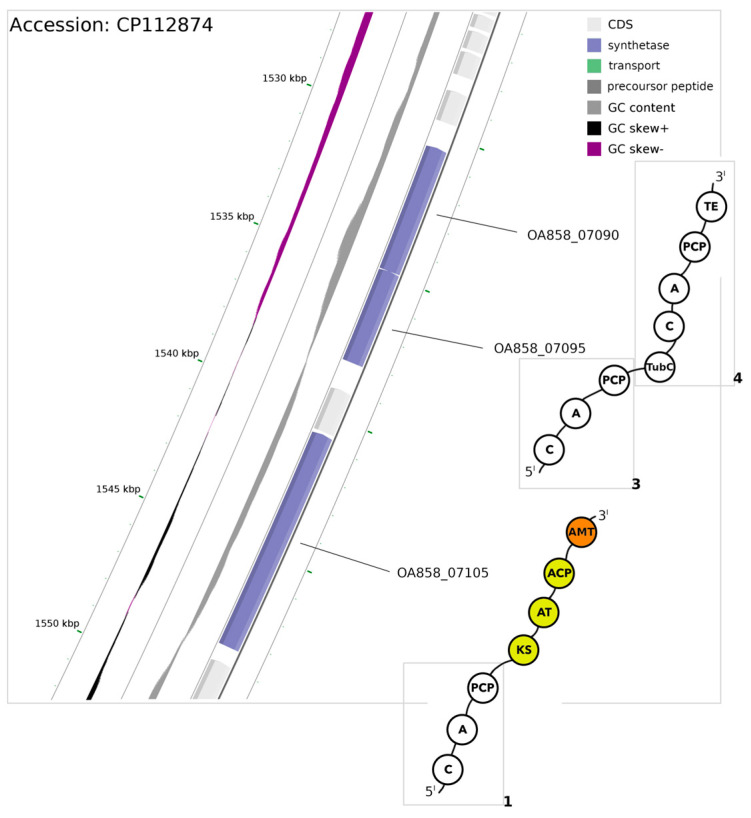
Closeup of the chromosome region encoding PKS/NRPS hybrid synthetase within a range of 1,530,477–1,548,363 bp encompassing three genes (marked blue) encoded on the reverse strand. Domains shown in circles are abbreviated as follows: A, adenylation; ACP, acyl carrier protein; C, condensation; KS, β-ketosynthase; AMT, aminotransferase; PCP, peptidyl carrier protein; TubC, N-terminal docking domain; TE, thioesterase. Colored domains represent the PKS fragment of synthetase. Domains in rectangles (1, 3, 4 in bold) correspond to the module numbering in Table 2.

**Figure 5 microorganisms-12-01628-f005:**
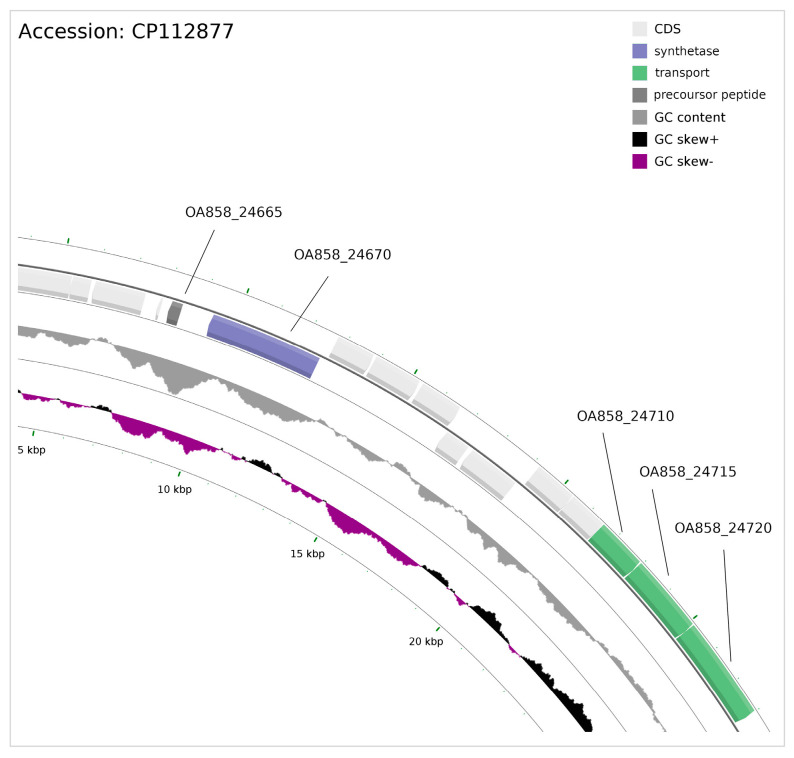
Closeup of the plasmid pPg_03 region encoding putative lanthipeptide synthetase, colored blue, with the precursor peptide, which is encoded upstream, colored dark gray. Downstream of the putative synthetase-encoding gene on the leading strand are three genes (green) related to transport and maturation of the precursor peptide (OA858_24665).

**Figure 6 microorganisms-12-01628-f006:**
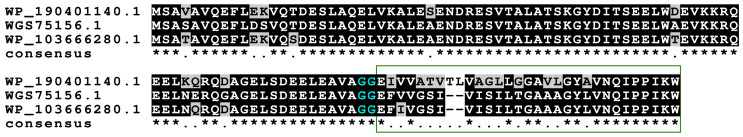
Alignment of RiPP precursor peptide (WGS75156.1) and two other precursors derived from *Pseudanabaena* strains. Downstream of the double glycine motif, a C-terminal core peptide is shown (marked by a green rectangle). Amino acids in gray indicate inconsistencies in sequence alignment. Asterisks indicate identical amino acid residues in all compared proteins.

**Table 1 microorganisms-12-01628-t001:** Amino acid residues in the substrate-binding pocket of the adenylation domains encoded by the NRPS gene (OA858_03905), according to GrsA Phe numbering.

Locus Tag	Proposed aa Activated	Residue Position According to GrsA Phe Numbering								
		235	236	239	278	299	301	322	330	331
OA858_03905 [module 1]	hydroxy/carboxy-acid	S	I	F	H	S	G	I	C	M
OA858_03905 [module 2]	Arg	D	A	E	D	I	G	T	V	V

**Table 2 microorganisms-12-01628-t002:** Amino acid residues in the substrate-binding pocket of the adenylation domains of NRPS/PKS hybrid synthetase, numbered in accordance with GrsA Phe.

Locus Tag	Proposed aa Activated	Residue Position According to GrsA Phe Numbering								
		235	236	239	278	299	301	322	330	331
OA858_07105 [module 1]	Val	D	A	L	W	L	G	G	T	F
OA858__07095 [module 3]	Val/Trp/Phe	D	A	F	W	L	A	G	T	F
OA858__07090 [module 4]	Thr/Val	D	V	E	N	I	G	G	V	T

**Table 3 microorganisms-12-01628-t003:** Peptides searched within coding sequences of *P. galeata* CCNP1313, found by Ceglowska et al. [18], where IDs in the Pattern column correspond with published peptides. TM in the Topology column stands for transmembrane, and inside and outside refer to the peptide’s position in cell. In the Pattern column, LI in brackets indicates regular expression applied in leucine or isoleucine positions, which cannot be differentiated via MS analysis.

Protein Sequence ID	Hypothetical Product	Topology	Pattern Name	Pattern	Strand	Start	End	Matched
WGS74979.1	sulfate permease (plasmid)	TM (inside)	PG638	RMGF[LI]	+	124	128	RMGFL
WGS75066.1	phosphate ABC transporter permease PstA (plasmid)	TM (helix)	GP598	A[LI]V[LI][LI]A	−	18	23	AIVLIA
WGS74721.1	AAA family ATPase (plasmid)	globular	GP598	A[LI]V[LI][LI]A	−	1066	1071	ALVLLA
WGS72582.1	solanesyl diphosphate synthase	globular	GP598	A[LI]V[LI][LI]A	+	55	60	ALVLLA
WGS73177.1	glycosyltransferase family 2 protein	TM (inside)	PG638	RMGF[LI]	+	325	329	RMGFL
WGS73241.1	hypothetical protein	globular	GP655	A[LI]V[LI][LI]AG	−	99	105	AIVILAG
			GP598	A[LI]V[LI][LI]A	−	100	105	AIVILA
WGS74578.1	glutamine-hydrolyzing carbamoyl-phosphate synthase small subunit	globular	GP598	A[LI]V[LI][LI]A	−	7	12	ALVLIA
WGS74238.1	DNA-directed RNA polymerase subunit gamma	globular	PG638	RMGF[LI]	+	101	105	RMGFI
WGS70420.1	hypothetical protein	globular	GP598	A[LI]V[LI][LI]A	−	28	33	AIVIIA
WGS71453.1	cation-translocating P-type ATPase	TM (inside)	GP655	A[LI]V[LI][LI]AG	−	435	441	ALVLLAG
			GP598	A[LI]V[LI][LI]A	−	436	441	ALVLLA
WGS71843.1	photosystem I core protein PsaB	TM (outside)	GP767	YAAF[LI][LI]A	+	729	735	YAAFLIA

## Data Availability

The sequence of the whole genome of *P. galeata* CCNP1313 is available at GenBank (BioProject accession number PRJNA884271). The MS results were deposited in the CyanoMetDB database (https://zenodo.org/records/7922070; last accessed on 7 August 2024), and are available in the Appendix A of this article, cited in this work as ref. [18]. The original contributions presented in the study are included in the article; further inquiries can be directed to the corresponding author.

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
