# Peer review of "Genome Mining of Pseudanabaena galeata CCNP1313 Indicates a New Scope in the Search for Antiproliferative and Antiviral Agents"

_microorganisms, 2024, doi:10.3390/microorganisms12081628_

Round 1

Reviewer 1 Report

Comments and Suggestions for Authors

The authors have performed a genome mining study of a specific cyanobacterial strain affiliated to Pseudoanabaena galeta, isolated from Gulf of Gdánsk, with the purpose to screen for novel antiproliferate and antiviral agents, since a former study had revelaed that 14 identified peptides did not show any antiproliferate/antiviral activities.  The genome mining study revealed four classes of newly described synthetases. The structure of these novel genes was elucidated - and shown to contain certain similarities to other potent bioactive compounds, however these could not be detected by MS - which may reflect some issues with current methodologies. 

A general comment on the  activity of the 14 tested peptides as mentioned in lines 12 and 71-72, maybe the activity cannot be attributed to isolated peptides (tested singly) but to a suitable combination of the peptides (or with some of the new candidates that you have identified through your imn silico genome study?

A general comment on your study: Could the issue with identifying potent antiproliferative and antiviral agents in your strain be related to the cyanophages, hindering the desired acitities? This is just a speculative question, but maybe it could be useful to check for the presence of cyanophage gene sequences to shed a further light on this issue. 

In your concluding remarks  you address methodological issues, have you got any suggestions for how this can be improved? 

Comments on the Quality of English Language

The manuscript is clearly written, however there are a few minor linguistic issues, e.g. an "a" is probably missing in e.g. the lines 48, 56 (cyanobacterial, cyanobactins), please check for similar errors in the rest of the manuscript.

Further, the authors may wish to consider if a "the" should be added to e.g. the lines 26 (located on one of the..)  and 37 (The majority...).

Author Response

REVIEWER’S COMMENT 1:

A general comment on the  activity of the 14 tested peptides as mentioned in lines 12 and 71-72, maybe the activity cannot be attributed to isolated peptides (tested singly) but to a suitable combination of the peptides (or with some of the new candidates that you have identified through your in silico genome study?

RESPONSE:

We thank the reviewer for this comment. We have incorporated this interesting idea into the discussion. The new text reads as follows (lines 524-529):

“Another intriguing phenomenon is that among 14 peptides isolated from P. galeata CCNP1313, none were active. One possibility is that the activity cannot be attributed to isolated peptides (tested alone) but rather to a suitable combination of the peptides. If this is true, one might even speculate that some of candidates for such compounds acting to-gether with others could belong to the group of molecules identified in this work on the basis of the in silico genome analysis.”

REVIEWER’S COMMENT 2:

A general comment on your study: Could the issue with identifying potent antiproliferative and antiviral agents in your strain be related to the cyanophages, hindering the desired acitities? This is just a speculative question, but maybe it could be useful to check for the presence of cyanophage gene sequences to shed a further light on this issue.

RESPONSE:

W agree that this was an attractive hypothesis. We have followed the idea and checked the presence of prophages in the genome of the investigated strain. However, the analysis indicated that the hypothesis is rather unlikely. The newly introduced description of the analysis is included in lines 476-493:

“3.8. Cyanophage DNA sequence in the P. galeata CCNP1313 genome

We have taken into consideration a possibility that the issue with identifying potent antiproliferative and antiviral agents in the P. galeata CCNP1313 strain might be related to the cyanophages, particularly prophages present in the cyanobacterial genome, which could encode proteins hindering the desired activities. Therefore, we have analysed the genome of the investigated strain for the presence of prophage-derived DNA sequences. Indeed, using PHASTER, a phage search tool [76], we have identified prophage incom-plete sequences within the genome. Putative prophage sequences were found within the chromosome, between 707,842 bp – 727,624 bp nucleotide span. This region was homol-ogous to several phages of which Planktothrix phage PaV-LD (NCBI: NC_016564.1), a cyanophage with 95,299 bp long genome, held the highest number (3) common genes. Out of 13 phage-related genes within the respective 19,783 bp span (21 genes in total), an en-donuclease-, transposase- and protease-encoding genes have been found. Rest of the phage-like genes were deemed phage-like proteins, or hypothetical proteins. No morpho-genesis related genes were found. We conclude that the region contains insufficient in-formation to elucidate on cyanophage activity within the P. galeata CCNP1313 genome. Therefore, the hypothesis on the interferences of phage-derived products with the activities of identified or predicted compounds seems to be unlikely.”

REVIEWER’S COMMENT 3:

In your concluding remarks  you address methodological issues, have you got any suggestions for how this can be improved? 

RESPONSE:

As recommended by the reviewer, we have suggested methodological improvements for further studies. The new text (lines 520-523) reads as follows:

“Therefore, one might propose that in further studies it is necessary to use conditions which ensure higher stability of tested compound or their protection against destabilizing factors. Moreover, using an MS system with a higher range of molecular weights of mole-cules which can be analysed may be recommended.“

 REVIEWER’S COMMENT 4:

Comments on the Quality of English Language

The manuscript is clearly written, however there are a few minor linguistic issues, e.g. an "a" is probably missing in e.g. the lines 48, 56 (cyanobacterial, cyanobactins), please check for similar errors in the rest of the manuscript.

Further, the authors may wish to consider if a "the" should be added to e.g. the lines 26 (located on one of the..)  and 37 (The majority...).

RESPONSE:

We thank for putting our attention to these errors. They were corrected, as were other typographical errors found in the manuscript.

Reviewer 2 Report

Comments and Suggestions for Authors

The manuscript is dedicated to the study of cyanobacteria genome and mining in it the biosynthetic clusters. The study is methodologically well, with all method well described. The structural analysis of the found genes is interesting and well described. Some comments:

1. It is not clear, why the authors resequenced the well studied strain?

2. The availability of the data both NGS and MS is not clear, the reference to publicly available datasets should be provided.

3. The connection between structural analysis and the MS analisys is not clear. Has been the production of predicted metabolites validated?

4. It is not clear, the described clusters are all found clusters? The whole statistics of the annotation should be provided.

Author Response

REVIEWER’S COMMENT 1:

  1. It is not clear, why the authors resequenced the well studied strain?

RESPONSE

To our knowledge, the genome of Pseudanabaena galeata CCNP1313 was not analysed previously. Before depositing the genome of this strain in the NCBI database by us, there were only single DNA fragments of this genome reported in the literature, like 16S rDNA sequence (GenBank accession number MN273769). Currently, there are only two Pseudanabaena galeata complete genome sequences deposited within NCBI database. The genome of the CCNP1313 strain, studied in this work (NCBI RefSeq assembly GCF_029910235.1) was deposited by us prior to Pseudanabaena galeata UHCC 0370 strain (NCBI RefSeq assembly GCF_034931695.1), and it is used as a reference genome. Note that CCNP1313 and UHCC0370 are two different strains of Pseudanabaena galeata.

REVIEWER’S COMMENT 2:

  1. The availability of the data both NGS and MS is not clear, the reference to publicly available datasets should be provided.

RESPONSE:

We are sorry we provided a wrong accession number to the Pseudanabaena galeata CCNP1313 genome (the previously indicated no. MN273769 is for 16S rDNA sequence of this strain, not the whole genome). Now, the proper accession number (GenBank BioProject PRJNA884271) is provided. Regarding the MS data, the results were deposited in the CyanoMetDB database  (https://zenodo.org/records/7922070). Moreover, detailed raw results are provided in the supplementary data to the article cited in our manuscript as ref. [18]. This information is provided now (lines 537-540)

REVIEWER’S COMMENT 3:

  1. The connection between structural analysis and the MS analisys is not clear. Has been the production of predicted metabolites validated?

RESPONSE:

We have recognized this limitation of the study. In fact, NMR analyses of the compounds were not performed due to low amounts of the material (thus, functional tests were chosen). This is also one of reasons why we decided to focus on in silico analyses. Another point is that some compounds may be synthesized only under specific environmental conditions, thus, despite the presence of corresponding genes, one cannot detect the presence or activity of specific molecules because they are either not produced or synthesized in very low amounts under standard growth conditions.

REVIEWER’S COMMENT 4:

  1. It is not clear, the described clusters are all found clusters? The whole statistics of the annotation should be provided.

RESPONSE

The whole statistics of the genomic analysis is available in the database (GenBank BioProject PRJNA884271), thus, we did not included it into this manuscript. Nevertheless, below we present some crucial numbers:

Assembly Method: Flye v. 2.9

Assembly Name: Pseudanabaena galeata CCNP1313

Genome Representation: Full

Genome Coverage: 456.0x

Sequencing Technology: Illumina NovaSeq; Oxford Nanopore GridION

Annotation Method: Best-placed reference protein set; GeneMarkS-2+

Annotation Software revision: 6.3

Features Annotated: Gene; CDS; rRNA; tRNA; ncRNA

Genes (total): 5,320

CDSs (total): 5,266

Genes (coding): 5,176

CDSs (with protein): 5,176

Genes (RNA): 54

rRNAs: 2, 2, 2 (5S, 16S, 23S)

complete rRNAs: 2, 2, 2 (5S, 16S, 23S)

tRNAs: 44

ncRNAs: 4

Pseudo Genes (total): 90

CDSs (without protein): 90

Pseudo Genes (ambiguous residues): 0 of 90

Pseudo Genes (frameshifts): 25 of 90

Pseudo Genes (incompl.): 66 of 90

Pseudo Genes (interal stop): 10 of 90

Pseudo Genes (multiple problems): 11 of 90

CRISPR Arrays: 6

Reviewer 3 Report

Comments and Suggestions for Authors

The reviewed MS is dedicated to studying Pseudanabaena galeata CCNP1313 genome for indications of antiproliferative and antiviral substances. The results of the paper are very important for microbiology, biotechnology of microorganisms, and pharmacology. The MS is clear and well-structured. The authors used proper methodology, which includes molecular-genetic analysis and using a public database. The methods are presented in detail, which makes results reproducible. The obtained results were carefully explained and illustrated by figures and tables. The data were interpreted consistently throughout the manuscript. Conclusions based on the results and well-reasoned.

After making some corrections, I recommend MS for publication.

Suggestion to the authors:

Major comments:

  1. I recommend you add keywords because it is important to paper recognition.
  2. Please add recent publications (within the last five years) to the reference list.
  3. For data availability statements, it is better to use the following the following sentence: The original contributions presented in the study are included in the article; further inquiries can be directed to the corresponding author.

Minor comments:

Line 21: Add the author name of Pseudanabaena galeata.

Line 75: Correct “an antiviral” to “and antiviral.”.

Figure 1, 2, 5: Please try to improve the visualization of figures and labels.

Line 183, 358, 460, 490: Add the author name of species and genus.

Line 434: P. galeata should be italicized.

Author Response

COMMENT 1: I recommend you add keywords because it is important to paper recognition.

RESPONSE 1: Keywords are included.

COMMENT 2: Please add recent publications (within the last five years) to the reference list.

RESPONSE 2: The manuscript cites 18 articles published within the last five years, which include all the most important papers in the field.

COMMENT 3: For data availability statements, it is better to use the following the following sentence: The original contributions presented in the study are included in the article; further inquiries can be directed to the corresponding author.

RESPONSE 3: The data availability statement has been complemented with the text suggested by the reviewer. 

COMMENT 4: Line 21: Add the author name of Pseudanabaena galeata.

RESPONSE 4: Added, as requested.

COMMENT 5: Line 75: Correct “an antiviral” to “and antiviral.”.

RESPONSE 5: Corrected, as requested.

COMMENT 6: Figure 1, 2, 5: Please try to improve the visualization of figures and labels.

RESPONSE 6: The Figures are supplied as high resolution images in the system.

COMMENT 7: Line 183, 358, 460, 490: Add the author name of species and genus.

RESPONSE 7: Names were added to taxonomic names when first appeared in the text.